# CodeLutra: Boosting LLM Code Generation via Preference-Guided Refinement

## Abstract

Large Language Models (LLMs) have significantly advanced code generation but often require substantial resources and tend to over-generalize, limiting their efficiency for specific tasks. Fine-tuning smaller, open-source LLMs presents a viable alternative; however, it typically lags behind cutting-edge models due to supervised fine-tuning's reliance solely on correct code examples, which restricts the model's ability to learn from its own mistakes and adapt to diverse programming challenges. To bridge this gap, we introduce CodeLutra, a novel framework that enhances low-performing LLMs by leveraging both successful and failed code generation attempts. Unlike conventional supervised fine-tuning, CodeLutra employs an iterative preference-guided refinement mechanism to compare correct and incorrect solutions as well as maximize the likelihood of correct codes. Through continuous refinement, CodeLutra enables smaller LLMs to match or surpass GPT-4's performance in various code generation tasks without relying on vast external datasets or larger auxiliary models. On a challenging data science coding task, using just 500 samples improved Llama-3-8B's accuracy from 28.2% to 48.6%, approaching GPT-4's performance. These results highlight CodeLutra's potential to close the gap between open-source and closed-source models, making it a promising approach in the field of code generation.

## 1 Introduction

Large language models (LLMs) have revolutionized numerous domains, consistently delivering great performance across different tasks (Brown et al., 2020; Achiam et al., 2023; Bubeck et al., 2023; Team et al., 2023; Anthropic, 2023; OpenAI, 2023). Among these applications, code generation stands out as particularly promising. Models pre-trained on extensive code repositories have demonstrated an impressive capability to solve diverse programming challenges (Li et al., 2023; 2022; Nijkamp et al., 2023; Zheng et al., 2023; Fried et al., 2022; Chen et al., 2021b; Wang et al., 2021b; 2023b).

Despite the promise, deploying ultra-large models for code generation poses significant challenges. Top-tier closed-source models like GPT-4 (OpenAI, 2023) are often highly resource-intensive and do not offer the flexibility for customization to specific code generation tasks. Fine-tuning smaller, open-sourced LLMs for targeted applications serves as a compelling alternative. Code generation demands not just syntactical correctness but also a deep understanding of logical and domain-specific nuances, which complicates model training. While a straightforward approach is to perform supervised fine-tuning (SFT) on the target task, this typically results in only incremental improvements and frequently lags behind cutting-edge solutions. This limited improvement stems from SFT's reliance solely on correct code examples, which restricts the model's ability to learn from its own mistakes and adapt to diverse programming challenges. We illustrate this in Figure 1, where SFT provides a minor boost in performance on the challenging data science task, with LlaMa-3-8B's Pass@1 improving from 28.2% to 30.0%. A notable gap remains between the fine-tuned model and GPT-4, which leads with a Pass@1 score of 49.4%. To address the issue, prior solutions collect additional training data by leveraging more powerful LLMs (Shen et al., 2023; Luo et al., 2023; Wang et al., 2023b; Yang et al., 2024) or by collecting external datasets from public repositories (Lozhkov

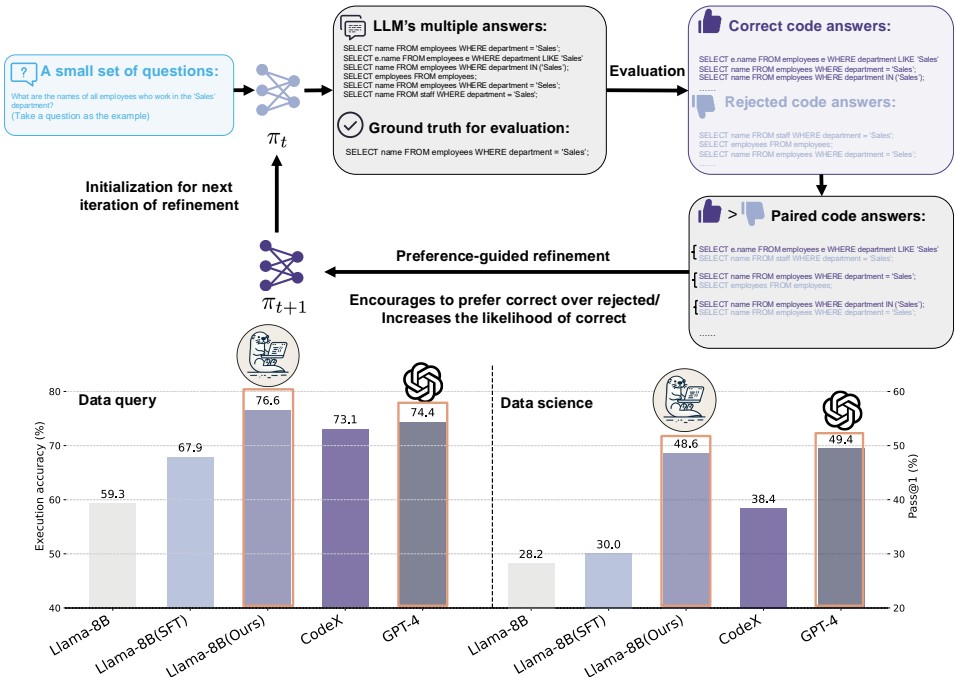

Figure 1: The proposed CODELUTRA framework (see Section 3) and performance comparison on different code generation tasks. The orange dashed box highlights our framework's performance relative to GPT-4.

et al., 2024; Muennighoff et al., 2023). *But the big elephant in the room still remains—how much can we bridge the gap by maximizing the utility of existing data and model at hand?*

To address this challenge, we introduce CODELUTRA, a framework that iteratively improves the performance of a given LLM *without relying on vast external datasets or larger auxiliary models*. Our approach demonstrates that even with limited data at hand, substantial gains in code generation quality can be achieved, closing the gap between smaller fine-tuned models and the top-tier LLMs. Unlike traditional fine-tuning methods that rely solely on correct code solutions, CODELUTRA constructs and learns from both successful and failed code generated by the current model, which forms self-generated comparative data. The failed code attempts are invaluable for refining models, since they provide concrete examples of common errors and enable the model to learn strategies for avoiding similar mistakes in future generations. To harness the comparative data, a key innovation of CODELUTRA is the preference-guided refinement mechanism that compares correct and incorrect code snippets, iteratively refining the model's understanding of code quality. With each iteration, the model generates code solutions, evaluates their correctness, and updates its parameters based on the evolving preference dataset. This process allows for continuous improvements of the base model, making the framework effective even with limited initial data (e.g., a few hundred samples).

We comprehensively evaluate the effectiveness of CODELUTRA on challenging data query and data science tasks, where the LLM is tasked with generating the correct SQL or Python code to solve a given problem. We compare CODELUTRA with 13 open-source and closed-source LLMs that are competitive in code generation. Notably, on the data query task, *our framework allows Llama-3-8B (Dubey et al., 2024) to achieve an execution accuracy of 76.6%, which exceeds GPT-4's 74.4%.* Under a challenging data science task, we find that using just 500 samples improved Llama-3-8B from an accuracy of 28.2% to 48.6%, approaching the performance of GPT-4. This demonstrates that CODELUTRA achieves strong results even with a limited number of high-quality annotations. Moreover, we observe the consistent performance gains of CODELUTRA on different base models, including Gemma-7B (Team et al., 2024) and StarCoder-7B (Dai & Kumar, 2023). These findings highlight the potential of CODELUTRA in closing the performance gap between open-source and closed-source models. To summarize our key contributions:

1. We present CODELUTRA, a novel framework that iteratively improves LLMs for code generation using self-generated comparative data from both successful and failed code at-

tempts, enabling low-performing models to rival top-tier solutions without external datasets or ultra-large LLMs' feedback.

2. We conduct comprehensive evaluations, comparing CODELUTRA against 13 competitive LLMs specializing in code generation. Results demonstrate CODELUTRA consistently outperforms both standard fine-tuned LLMs and existing cutting-edge closed-source LLMs.

3. We conduct a series of in-depth analyses to understand the contribution of failed attempts and likelihood regularization for CODELUTRA, and reveal CODELUTRA improves the performance via reducing syntax errors and improving incorrect answers across iterations.

## 2 PRELIMINARIES

**LLMs for code generation.**  LLMs are pre-trained on diverse datasets encompassing both natural and programming languages. In code generation tasks, an LLM receives a prompt—such as a natural language description, and generates the corresponding code by predicting the next token in the sequence. Formally, code generation is modeled as the conditional probability of a code sequence $y = (y_1, y_2, \ldots, y_T)$ given an input prompt $x$:

$$P(y|x) = \prod_{t=1}^{T} P(y_t|y_{<t}, x) \tag{1}$$

where $x$ is the input prompt, $y$ is the generated code sequence of length $T$, and $y_{<t} = (y_1, y_2, \ldots, y_{t-1})$ represents the tokens generated before time step $t$.

**Supervised fine-tuning on task-specific dataset.**  Pre-trained LLMs can be suboptimal on task-specific dataset, necessitating fine-tuning. We consider a task-specific dataset $\mathcal{D} = \{(x_i, y_i)\}_{i=1}^{n}$ containing $n$ examples, where each pair $(x_i, y_i)$ represents an input prompt $x_i$ and its corresponding target code $y_i$. Supervised fine-tuning (SFT) adjusts the model's parameters $\theta$ by maximizing the likelihood of generating correct code sequences $y_i$. The loss is defined as:

$$\pi_{\text{SFT}} = \text{argmax}_{\pi_\theta} \mathbb{E}_{(x_i, y_i) \sim \mathcal{D}} \left( \log \pi_\theta \left( y_i | x_i \right) \right). \tag{2}$$

This process relies on the quality of the dataset, guiding the model to produce more accurate and reliable code based on the given prompts.

**Verification of code correctness.**  We verify the correctness of the generated code by comparing it with the ground truth code $y_{\text{gt}}$. We define a verification function $V(y)$ that checks if the output of the generated code matches that of the ground truth. Given an input $\mathbf{I}$, the generated code is correct if it produces the same output $f(y, \mathbf{I})$ as the ground truth code $f(y, \mathbf{I})$. Specifically, $V(y) = 1$ if the generated code is correct, and $V(y) = 0$ otherwise.

**Limitations of SFT for code generation.**  Code generation demands not just syntactical correctness but also a deep understanding of logical and domain-specific nuances, which complicates model training. *A major limitation of SFT is that it solely maximizes the likelihood of providing correct code, which restricts the model's ability to learn from its own mistakes*. Since the training process focuses exclusively on provided examples, the LLM doesn't receive the gradient from incorrect or suboptimal code. For instance, if the model only predicts wrongly in the final token in a code snippet, the overall probability $P(y|x)$ in the Equation 1 might still remain high as the preceding tokens are correct. Correspondingly, the SFT loss is very small in that case. However, this single erroneous token can render the entire code nonfunctional or introduce subtle bugs, significantly affecting execution correctness despite a high likelihood score, which is different from general NLP tasks. This reliance on only correct examples limits the model's capacity to identify and recover from errors, reducing its ability to handle more complex or nuanced coding tasks. This motivates our framework CODELUTRA, which leverages both successful and failed code generation attempts.

## 3 CODELUTRA

In this section, we introduce CODELUTRA, a framework designed to comparatively learn from both correct and incorrect code generations. CODELUTRA delivers substantial performance gains,

achieving performance comparable to more advanced models like GPT-4 (OpenAI, 2023), even with limited initial data. The pseudo-code is provided in the Appendix A.1.

**Initialization.** We start with an initial base model, denoted as $\pi_0$, which serves as the starting point for our iterative refinement process. We are provided with an initial training set $\mathcal{D} = \{(x_i, y_i)\}_{i=1}^{n}$, where each $x_i$ is a natural language query, and $y_i$ is the corresponding ground truth code solution. Starting with a modestly performing model allows us to clearly observe improvements attributable to the CODELUTRA framework, ensuring that enhancements result from our methodology rather than inherent model capabilities. Note that at initialization, we only have the correct codes in hand. We describe next how to obtain incorrect codes to serve the model refinement.

**Generating correct and failed code.** At each iteration $t$, the current model $\pi_t$ generates multiple code responses for each input query $x_i \in \mathcal{D}$. Specifically, for each $x_i$, the model produces $M$ distinct code samples:

$$\hat{y}_i^m \sim \pi_t(x_i), \quad \text{for each } m \in \{1, 2, \ldots, M\}. \tag{3}$$

Generating multiple responses introduces diversity in the model's outputs, providing a richer dataset for composing preference datasets. Each generated code snippet $\hat{y}_i^m$ is then executed to assess its correctness by comparing the execution result against that of the ground truth solution $y_i$. Correct executions are categorized into the correct code set $Y_i^{(c)}$, while incorrect ones are placed into the rejected set $Y_i^{(r)}$. This evaluation mechanism offers clear feedback, particularly syntax and execution errors common in code generation tasks, and enables the construction of preference dataset.

**Preference dataset construction.** A key of our framework involves constructing a preference dataset $\mathcal{D}_t$ that captures the relative quality of generated code. The preference data is updated at every iteration $t$. For each input $x_i$, we create $K$ preference pairs by randomly pairing one correct code $\hat{y}_i^{c_k} \in Y_i^{(c)}$ with one rejected code $\hat{y}_i^{r_k} \in Y_i^{(r)}$:

$$(\hat{y}_i^{c_k}, \hat{y}_i^{r_k}), \quad \text{for each } k \in [K]. \tag{4}$$

If either $Y_i^{(c)}$ or $Y_i^{(r)}$ contains fewer than $K$ responses, sampling with replacement is employed to maintain consistency. The complete preference dataset at iteration $t$ is thus defined as:

$$\mathcal{D}_t = \{(x_i, \hat{y}_i^{c_k}, \hat{y}_i^{r_k}) \mid \text{ for all } x_i \in \mathcal{D} \text{ and } k \in [K]\}, \tag{5}$$

which contains $n \times K$ preference triplets. Here, $n$ corresponds to the size of the initial dataset $\mathcal{D}$, which remains constant throughout iterations. This dataset encapsulates nuanced comparisons between correct and incorrect code generations, facilitating the subsequent preference learning step. By systematically pairing correct and rejected responses, the model gains a clearer understanding of high-quality code, enabling targeted improvements.

**Preference-guided refinement.** While one can directly employ a preference optimization approach like DPO (Rafailov et al., 2023) on our curated dataset $\mathcal{D}_t$, this approach presents a notable limitation due to its tendency to decrease the likelihood of both correct and rejected code during training. This is evidenced by the dashed lines in Figure 2, which is also observed in Pal et al. (2024); Feng et al. (2024). This diminishing likelihood can significantly impact our framework, especially since our correct code are critical for successfully solving the assigned tasks. It is crucial, therefore, to prioritize the likelihood of correct solutions to the task at hand.

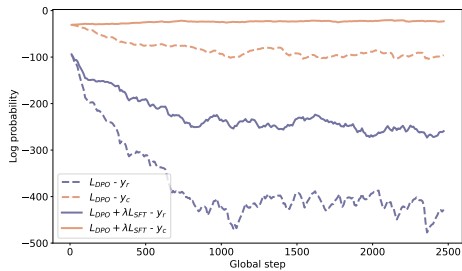

Figure 2: Effect of SFT loss to keep the likelihood of correct answers stable.

To address this limitation, we employ a dual-loss function integrating DPO with SFT, which regularizes the training to prevent decreasing likelihood. Specifically, we employ SFT on the dataset with correct solutions, $\mathcal{D}_t^c = \{(x_i, \hat{y}_i^{c_k}) \mid \text{ for all } x_i \in \mathcal{D} \text{ and } k \in [K]\}$. The overall loss function is

defined as:

$$\pi_{t+1} = \text{argmin}_{\pi_\theta} \left[ \underbrace{-\mathbb{E}_{(x_i, y_i^c, y_i^r) \sim \mathcal{D}_t} \left[ \log \sigma \left( \beta \left( \log \frac{\pi_\theta(y_i^c|x_i)}{\pi_t(y_i^c|x_i)} - \log \frac{\pi_\theta(y_i^r|x_i)}{\pi_t(y_i^r|x_i)} \right) \right) \right]}_{\text{compare correct and incorrect solutions}} \right.$$

$$\left. \underbrace{- \lambda \mathbb{E}_{(x_i, y_i^c) \sim \mathcal{D}_t} \left( \log \pi_\theta \left( y_i^c|x_i \right) \right)}_{\text{maximize likelihood for correct codes}} \right], \tag{6}$$

where $\lambda$ is a hyperparameter balancing the contributions of the DPO and SFT losses. The first term facilitates preference-based fine-tuning by optimizing the model to favor correct over incorrect code. Concurrently, the second term enhances the likelihood of generating correct solutions directly to avoid the log probability decreasing (see solid line in Figure 2). This dual-loss approach ensures that the model not only ranks correct solutions higher but also increases their generation probability, leading to more consistent high-quality code outputs. We verify the effectiveness of the dual loss empirically in Section 4.3. The refinement process continues until the improvement between consecutive iterations $\pi_t$ and $\pi_{t+1}$ becomes marginal, indicating convergence.

**Remark.** While DPO-style preference optimization has been studied in recent literature (see review in Section 6), its connection to code generation remains largely unexplored. To the best of our knowledge, our work is the first to establish this critical link between iterative preference learning and code generation. We highlight several novel aspects that differentiate our approach from prior work. First, previous studies mostly focus on natural language generation tasks and rely on model-generated rewards or feedback from other large models (Chen et al., 2024; Xiong et al., 2024; Yuan et al., 2024; Pang et al., 2024; Xie et al., 2024). In contrast, we focus exclusively on code generation where the preference labels for iterative refinement come from execution results. This shift is essential and unique for code generation, where execution correctness is a key indicator of preference. Moreover, our work uncovers the insight into the dual-loss mechanism, revealing that it plays a crucial role in improving performance in the context of code generation. From a practical standpoint, our training pipeline bypasses the standalone SFT stage typically required before the preference optimization phase, thereby streamlining the training process. Overall, our work not only establishes a novel link between iterative preference learning and code generation but also introduces practical innovations that enhance performance, marking a significant step forward in the field.

## 4 EXPERIMENTS

### 4.1 EXPERIMENTAL SETUP

**Tasks.** To evaluate the effectiveness of our proposed framework, we perform experiments on two tasks: **Data Query** and **Data Science**. Both tasks reflect common and practical challenges in fields such as business, healthcare, and scientific computing, where precise code generation is critical for solving data-related problems while receiving limited attention. For the Data Query task, the model is given a natural language problem description and is tasked with generating the corresponding SQL query for a database using an LLM. For example, given the description "*How many heads of the departments are older than 56?*", the model should produce the appropriate SQL query to execute this request. In the Data Science task, the LLM is tasked with generating the correct Python code to solve a given data science problem. For instance, given the problem "*I have a 2D array to represent a many-many mapping. What is the quickest way to zero out the second row and the first column?*", we test LLM's ability to solve data science problems with `numpy`. We provide examples of the two tasks of the question and ground truth code in the Appendix A.2

**Datasets.** We conduct our experiments on two cross-domain datasets for data query, *Spider* (Yu et al., 2018) and *BIRD* (Wang et al., 2023a), as well as a data science dataset, *DS-1000* (Lai et al., 2023). *Spider* includes 10,181 questions with 5,693 unique SQL queries across 200 databases in 138 domains, while *BIRD* contains 12,751 question-SQL pairs across 95 large databases, covering over 37 domains. We utilize *DS-1000*, which comprises 1,000 data science problems sourced from Stack Overflow, covering seven Python libraries related to analysis in data science. The dataset is designed to minimize memorization risk by modifying original problems and uses a multi-criteria

Table 1: The Execution Accuracy (EX), Exact Match (EM), and Pass@1 for different kinds of models on SPIDER, BIRD, and DS1000. We show the base model ($\pi_0$) without fine-tuning and the model trained with CODELUTRA in different iteration ($\{\pi_1, \pi_2, \pi_3, \pi_4\}$). For fair comparison, all reported results in the table use the same prompt. **Boldface** highlight GPT-4 and our results.

| Models | Spider | | BIRD | | DS1000 |
|---|---|---|---|---|---|
| | EX | EM | EX | EM | Pass@1 |
| *Open-source LLMs* | | | | | |
| Llama-3-8B (Dubey et al., 2024) | 59.3 | 55.1 | 22.3 | 19.5 | 28.2 |
| Codellama-7B (Xu & Zhang, 2023) | 57.0 | 51.4 | 24.4 | 18.7 | 25.6 |
| StarCoder-7B (Lozhkov et al., 2024) | 61.2 | 58.6 | 25.7 | 23.0 | 26.8 |
| Gemma-7B (Team et al., 2024) | 49.9 | 46.7 | 21.2 | 19.1 | 24.2 |
| Codestral-22B (Brown & Lee, 2023) | 71.3 | 69.6 | 42.5 | 39.9 | 35.8 |
| Llama-3-70B-Instruct (Dubey et al., 2024) | 68.7 | 65.4 | 41.2 | 39.3 | 36.4 |
| *Fine-tuned LLMs* | | | | | |
| Llama-3-8B (Dubey et al., 2024) | 67.9 | 64.7 | 35.6 | 30.7 | 30.0 |
| Codellama-7B (Xu & Zhang, 2023) | 67.3 | 64.3 | 36.3 | 30.9 | 26.8 |
| StarCoder2-7B (Lozhkov et al., 2024) | 66.9 | 64.1 | 36.6 | 31.1 | 29.4 |
| Gemma-7B (Team et al., 2024) | 65.8 | 62.8 | 34.5 | 29.8 | 27.4 |
| *Closed-Source LLMs* | | | | | |
| Codex (Chen et al., 2021a) | 73.1 | 70.2 | 44.7 | 42.4 | 38.4 |
| ChatGPT (Ouyang et al., 2022) | 71.8 | 68.4 | 44.3 | 40.2 | 38.8 |
| GPT-4 (OpenAI, 2023) | **74.4** | **71.2** | **46.3** | **43.2** | **49.4** |
| CODELUTRA *(Ours)* | | | | | |
| Base ($\pi_0$) | 59.3 | 55.1 | 22.3 | 19.5 | 28.2 |
| Iteration 1 ($\pi_1$) | 67.8 | 63.9 | 37.8 | 33.2 | 43.2 |
| Iteration 2 ($\pi_2$) | 72.4 | 68.3 | 40.8 | 36.0 | 46.8 |
| Iteration 3 ($\pi_3$) | 76.6 | 72.5 | 43.1 | 38.6 | 48.6 |
| Iteration 4 ($\pi_4$) | **76.3** | **72.1** | **42.6** | **38.3** | **48.2** |

evaluation system to assess functional correctness and coding constraints. We split DS-1000 into 500 samples for training and 500 for evaluation.

**Metrics.** For the *Data Query* task, we adopt the metrics introduced by Yu et al. (2018): **Execution Accuracy (EX)**, which measures whether the SQL query execution result matches the expected output, and **Exact Match (EM)**, which evaluates whether the generated SQL query exactly matches the reference query in both structure and semantics. For the *Data Science* task, we use **pass@1**, following Lai et al. (2023), which indicates the percentage of correct solutions generated by the model on the first attempt.

**Baselines.** To evaluate the effectiveness of our method, we compare it against three categories of baselines. For a fair comparison, all reported results are based on the same prompt.

- *Open-source LLMs:* We benchmark our method against competitive open-source LLMs, including models pre-trained on general datasets such as **Llama-3-8B** (Dubey et al., 2024), **Gemma-7B** (Team et al., 2024), and **Llama-3-70B-Instruct** (Dubey et al., 2024). Additionally, we compare against LLMs pre-trained specifically on coding datasets, such as **Codellama-7B** (Xu & Zhang, 2023), **StarCoder-7B** (Lozhkov et al., 2024), and **Codestral-22B** (Brown & Lee, 2023).

- *Fine-tuned LLMs:* As supervised fine-tuning on domain-specific datasets is a popular and effective way to improve LLMs' corresponding performance, we also report the performance of fine-tuned LLMs using standard supervised fine-tuning methods (Raffel et al., 2020).

- *Closed-source LLMs:* We provide the performance of cutting-edge closed-source LLMs, including **Codex** (Chen et al., 2021a), **ChatGPT** (Ouyang et al., 2022), and **GPT-4** (OpenAI, 2023).

**Experimental setup.** For main results, we apply our framework to the **Llama-3-8B** base model (Dubey et al., 2024), denoted as $\pi_0$ (see Section 4.2 for more backbone results). We use a zero-shot prompt containing the question along with reference information (dataset schema for data query and reference code for data science). For different answer collections, we employ the best-of-$n$ strategy by sampling 16 responses at the temperature of 1.0. We train one epoch per iteration and perform four iterations in total, resulting in models $\{\pi_1, \pi_2, \pi_3, \pi_4\}$. These models are evaluated as described in the following sections. For more experimental details, please refer to the Appendix A.4.

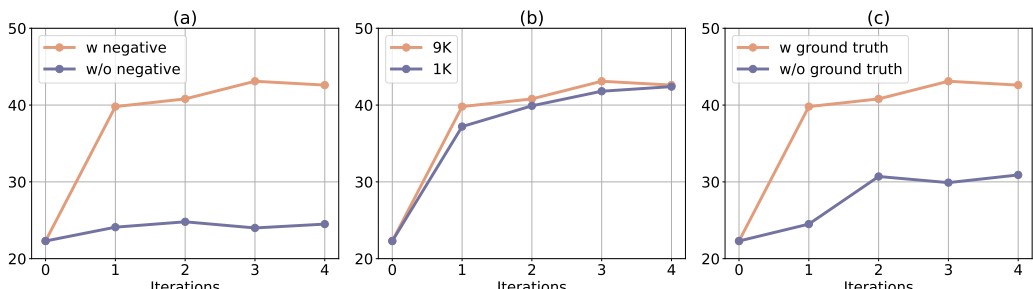

Figure 3: (**a**) Ablations on the effects of negative samples for training. (**b**) Ablations on the question number during training. (**c**) The effects of ground truth for validation during preference datasets collection.

## 4.2 MAIN RESULTS

**Results on the data query task.** We compare CODELUTRA with baselines in code generation for the data query task, as shown in Table 1. We found that existing open-source LLMs like Llama-3-8B still have a significant performance gap in code generation for data queries compared to closed-source LLMs like GPT-4. Although supervised fine-tuning can help bridge this gap—e.g., SFT increases the EX of Llama-3-8B on Spider from 59.3% to 67.9%—there remains a notable difference with GPT-4's 74.4%. Through our refinement framework, Llama-3-8B after four iterations exceeded SFT performance by 16.9% and *even outperforms GPT-4 with an execution accuracy of 76.6%*. Additionally, on the more challenging BIRD dataset, after three iterations, CODELUTRA significantly improved the EX of the base model from 22.3 to 43.1, achieving performance very close to GPT-4.

**Results on the data science task.** Table 1 also presents results for the data science task, where we evaluate both open-source and closed-source LLMs, as well as our method CODELUTRA. On the DS-1000 dataset, open-source models like Llama-3-8B and Gemma-7B struggle, with significantly lower EM and Pass@1 scores compared to closed-source models like GPT-4. Fine-tuning provides a minor boost in performance, as seen with Llama-3-8B's Pass@1 improving from 28.2% to 30.0%. However, as with the data query task, a large performance gap remains between fine-tuned open-source models and closed-source ones, where GPT-4 leads with a Pass@1 score of 49.4%. Nonetheless, CODELUTRA demonstrates substantial improvements (from 28.2% to **48.6%**), offering a promising path for narrowing this gap further.

## 4.3 MORE EXPERIMENTS

**The importance of learning from failed attempts.** Our framework CODELUTRA leverages both positive and negative answer pairs to iteratively improve model performance, particularly by minimizing the generation of incorrect responses. But what happens when we omit the negative samples and rely solely on supervised fine-tuning using positive samples generated by the model? In this ablation, we compare the performance of our objective 6 with a model trained with $\mathcal{L}_{\text{SFT}}(\pi; \mathcal{D}_t^c)$. As seen in Figure 3(a), without negative samples (purple line), the model's performance plateaus across iterations, remaining close to the baseline. In contrast, incorporating negative samples (blue line) leads to steady performance improvements over successive iterations. This ablation confirms that including negative samples is critical to refining the model's ability to distinguish between optimal and suboptimal responses, significantly boosting overall performance. Collecting incorrect answers is thus an essential component for preference learning and contributes to greater model accuracy.

**CodeLutra achieves strong performance under limited training data.** The cost of acquiring high-quality question-code pairs can be significant, so we examine whether our method truly depends on large datasets. Under the data science code generation task, *we found that using just 500 samples improved Llama-3-8B from an accuracy of 28.2 to 48.6, approaching the performance of GPT-4*. This demonstrates that CODELUTRA achieves strong results even with a limited number of high-quality annotations. We further verify this with the data query task by randomly selecting 1K question-code pairs from BIRD's training data and comparing them to the full 9K sample set. The results, as shown in Figure 3(b), reveal similar trends, with the two setups reaching peak execution

accuracies of 43.1 and 42.4, respectively. This minor difference suggests that CODELUTRA does not heavily rely on large volumes of training data and can generalize well with fewer annotations, which is crucial for minimizing the cost of dataset collection.

**Importance of SFT regularization during preference optimization.** Recall in Section 3 that our loss function integrates DPO with SFT to regularize the training, and prevent decreasing likelihood on the correct solution. In Table 1,we ablate the effect of SFT regularization on both the Spider and DS-1000 datasets. Notably, omitting optimizing the SFT loss on the correct solutions results in a marked decline in model performance, *e.g.*,

Table 2: Ablations on the SFT on the correct answers.

| Methods | Spider | DS1000 |
|---|---|---|
| $\mathcal{L}_{\text{DPO}}$ | 17.2 | 12.4 |
| Ours | **76.6** | **48.6** |

↓59.4% on Spider. This highlights the effectiveness of the dual loss approach, ensuring that the model not only ranks correct solutions higher but also increases their generation probability, leading to more consistent high-quality code outputs.

**CODELUTRA remains effective on different base models.** To further validate the generalization capability of our framework CODE-LUTRA, we extend our experiments to two additional open-source base models: **Gemma-7B** (Team et al., 2024) and **StarCoder-7B** (Lozhkov et al., 2024). As summarized in Table 3, we report the results on both the *Spider* and *DS1000* datasets across multiple iterations of our refinement process. For Gemma-7B, we observe a significant improvement in Execution Accuracy (EX) on Spider, starting from 49.9% at $\pi_0$ (the base model) and reaching 72.6% after

| Model | Gemma-7B | | StarCoder-7B | |
|---|---|---|---|---|
| | Spider | DS1000 | Spider | DS1000 |
| $\pi_0$ | 49.9 | 24.2 | 61.2 | 26.8 |
| $\pi_1$ | 63.7 | 38.8 | 72.8 | 39.6 |
| $\pi_2$ | 69.3 | 43.6 | 74.7 | 42.4 |
| $\pi_3$ | 71.3 | 44.4 | 77.2 | 45.2 |
| $\pi_4$ | **72.6** | **44.0** | **77.5** | **45.8** |

Table 3: Performance with CODELUTRA of Gemma-7B and StarCoder-7B across Spider and DS1000 benchmarks.

four iterations ($\pi_4$). A similar trend is observed in the DS1000 dataset, where the Pass@1 metric improves from 24.2% to 44.0%. For StarCoder-7B, the improvements are also pronounced, with EX on Spider increasing from 61.2% to 77.5%, and Pass@1 on DS1000 rising from 26.8% to 45.8%. These results demonstrate that our framework is robust across different model architectures, consistently yielding significant performance gains regardless of the underlying base model. Notably, the iterative refinement process of CODELUTRA continues to improve the accuracy and correctness of generated code, highlighting the CODELUTRA generalization to different code generation tasks.

---

**Key Takeaways from Section 4**

1. **Failed attempts matter**: Incorporating negative samples in training leads to strong performance improvements, while models trained only on positive samples plateau. CODELUTRA performs on par with or even outperforms GPT-4 on data query and data science tasks, closing the gap between open-sourced and closed-sourced models.

2. **Strong performance with limited data**: Our method achieves significant accuracy improvements even with small datasets (e.g., improving Llama-3-8B's accuracy from 28.2 to 48.6 with only 500 samples), demonstrating its effectiveness without reliance on large volumes of training data.

3. **Importance of likelihood regularization**: Ablations show that incorporating SFT alongside preference optimization is crucial, highlighting the necessity of our dual-loss approach for maintaining high-quality code outputs.

---

## 5 FURTHER ANALYSIS ON CODELUTRA

**Is ground truth code necessary for preference dataset collection?** Recall that our framework relies on ground truth code to evaluate the quality of generated code during the collection of preference datasets. To test the impact of this dependence, we conduct experiments that replace the ground truth with a more general criterion—whether the generated code is executable. In the absence of ground truth, we consider executable answers as chosen and non-executable ones as rejected. Applying this

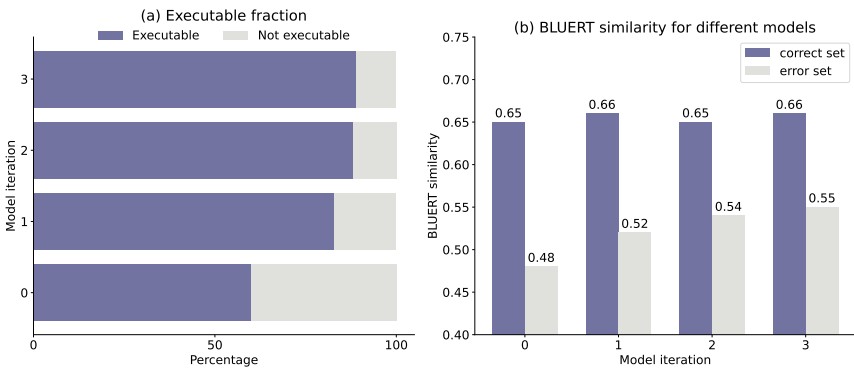

Figure 4: (**a**) Error analysis for different models. (**b**) BLEURT similarity for different models with the ground truth on the correct set and error set divided from the base model.

---

**Different answers from different models for the data query:**

**Database schema:** CREATE TABLE customers ( CustomerID INTEGER UNIQUE not null primary key, Segment TEXT null, Currency TEXT null ); CREATE TABLE "yearmonth" ( CustomerID INTEGER not null references customers on update cascade on delete cascade, Date TEXT not null, Consumption REAL, primary key (Date, CustomerID) );

**Question**: Among the customers who paid in euro, how many of them have a monthly consumption of over 1000?

**The answer from $\pi_0$ (wrong):**

```
SELECT COUNT(CustomerID) FROM yearmonth WHERE Consumption > 1000
AND CustomerID IN (
    SELECT CustomerID FROM customers WHERE Currency = 'EUR' );
```

*SQLite error: no such column: Currency*

**The answer from $\pi_3$ (correct):**

```
SELECT COUNT(CustomerID) AS "Number of customers" FROM "yearmonth"
WHERE  Consumption > 1000 AND   Currency = 'EUR'   GROUP BY CustomerID
```

---

Figure 5: CODELUTRA helps reduce the syntax errors. $\pi_0$ fails due to a missing column, while $\pi_3$ generates the correct query.

approach to the Bird dataset, we observe notable gains despite the absence of ground truth: accuracy rose from 22.3 to 30.9 (see Figure 3(c)). Moreover, the proportion of executable code surged from 59.8% to 89.7%, showing that the model effectively learned to avoid common errors, such as syntax issues or missing database tables. *This experiment demonstrates that using executability as a metric still enables substantial model improvements, making the method applicable even without high-quality annotations*, and highlights the robustness of CODELUTRA under such conditions.

**CODELUTRA helps reduce the syntax errors across iterations.** To evaluate whether our method enables LLMs to learn from their mistakes over multiple iterations, we sampled 100 error cases from the test set using models $\pi_0, \pi_1, \pi_2, \pi_3$, trained with CODELUTRA on the BIRD dataset for qualitative analysis. We measure the fraction of executable code generated by each model. As shown in Figure 4, the percentage of non-executable code decreases from 40% to 11% when trained with CODELUTRA, indicating that the models have improved in mastering SQL syntax and are better at avoiding basic errors. A qualitative example in Figure 5 highlights this improvement: the base model incorrectly queries the "Currency" column in the wrong table, resulting in an error, while the model trained with CODELUTRA successfully generates the correct SQL query.

**CODELUTRA improves quality of incorrect answers across iterations.** Based on the responses of the initial model $\pi_0$, we divide the test set into a correct set and an error set. We track the quality trends of the model in these two sets across iterations. Using the cosine similarity metric based on the BLEURT embedding proposed by Sellam et al. (2020), we calculate similarities denoted as $\text{sim}\pi_t(\hat{y}, y_{\text{gt}})$ for the model fine-tuned over $t$ iterations. Here $\hat{y}$ denotes the model generation, and

$y_{\text{gt}}$ is the ground truth solution. As shown in Figure 3(b), we observe that the similarity between the model's output on the correct set and the ground truth remains stable (see purple bars), while with each iteration, the similarity between the model's output on the error set and the ground truth increases significantly—from 0.48 to 0.54. This indicates that CODELUTRA helps the base model improve outputs on error set, while the outputs on correct cases remain qualitatively stable.

## 6  RELATED WORK

**Preference learning for LLMs.**  Preference learning aims to guide language models toward generating outputs that align with human preferences and desirable outcomes. A significant body of research has utilized human feedback to refine LLMs across various language tasks (Ziegler et al., 2019; Ouyang et al., 2022; Stiennon et al., 2020; Kreutzer et al., 2018). The Reinforcement Learning from Human Feedback (RLHF) framework, in particular, has been effective in aligning large pre-trained language models (Christiano et al., 2017; Ziegler et al., 2019; Ouyang et al., 2022; Bai et al., 2022). However, RLHF can suffer from training inefficiencies and sensitivity to hyperparameters. In response, recent studies have shifted towards closed-form loss functions that directly utilize offline preference data, exemplified by DPO (Rafailov et al., 2023) and related methodologies (Liu et al., 2023; Ethayarajh et al., 2024; Gheshlaghi Azar et al., 2024; Pal et al., 2024; Liu et al., 2023; Xiong et al., 2023a; Tang et al., 2024; Yu et al., 2024). While DPO-style models inherently provide rewards, iterative DPO—where the model generates its own pairwise preference data—has demonstrated strong performance and potential (Chen et al., 2024; Xiong et al., 2024; Yuan et al., 2024; Rosset et al., 2024; Pang et al., 2024; Xie et al., 2024). In this work, we introduce iterative preference-guided refinement to code generation for the first time, achieving GPT-4-level results and providing an in-depth analysis that paves the way for future research.

**LLMs for code generation.**  LLMs trained on vast corpora of code have demonstrated remarkable capabilities across a wide range of tasks, including code generation (Chen et al., 2021c; Austin et al., 2021; Zhang et al., 2022), program repair (Xia & Zhang, 2022; Wei et al., 2023; Xia et al., 2023; Jiang et al., 2023; Bouzenia et al., 2024; Xiong et al., 2023b), and software testing (Chen et al., 2023; Wang et al., 2024a; Zhou et al., 2024). Foundational models (Nijkamp et al., 2022; Wang et al., 2021a; Li et al., 2023; Roziere et al., 2023) pre-trained on extensive codebases, have established strong general-purpose capabilities for code generation. Building upon these powerful foundations, more recent models like WIZARDCODER (Luo et al., 2023) and DS-CODER (Li et al., 2023; Bouzenia et al., 2024) enhance contextual understanding by leveraging repository-level organization of pretraining data and incorporating retrieval-augmented techniques (Borzunov et al., 2024). Moreover, CODEINSTRUCT (Wang et al., 2024b) capitalize on instruction fine-tuning to improve alignment with human coding preferences. Fine-tuning methodologies and prompt-engineering techniques (Luo et al., 2023; Chen et al., 2023; Zhang et al., 2024b) have been crucial in unlocking these models' full potential for domain-specific tasks, such as security, AI-assisted development, and code synthesis in specialized fields. Zhang et al. (2024a) pay attention to preference learning for programming problems. However, they rely on GPT-4 for generating test cases and use preference data in a single offline run. In contrast, we *iteratively* refine a small LLM using self-generated preference data, *without external datasets or larger models*. Moreover, different from Zhang et al. (2024a) performing SFT and preference learning in two stages, we introduce the dual-loss to compare correct and incorrect solutions and maximize the likelihood of correct codes in one stage.

## 7  CONCLUSION

We introduced CODELUTRA, a preference-guided refinement framework designed to enhance LLMs for code generation without the need for external datasets or larger models. By utilizing self-generated data from both successful and failed attempts, CODELUTRA refines the model's understanding of code quality. Our experiments on data query and data science tasks demonstrate that CODELUTRA significantly boosts the performance of base LLMs. For example, fine-tuning Llama-3-8B with CODELUTRA outperforms GPT-4 on data query tasks and nearly matches GPT-4 performance on data science task with small training data. Additionally, CODELUTRA reduced common coding errors while improving the quality and accuracy of generated code. These results underline CODELUTRA's potential as a cost-efficient and scalable solution for LLM code generation.

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

## A   MORE INFORMATION

### A.1   PSEUDO CODE OF CODELUTRA

We summarize our algorithm in implementation in the Algorithm 1. The algorithm operates itera-
tively, generating code responses for each input and leveraging both correct and incorrect code sam-
ples to construct a preference dataset. These comparisons serve as the basis for model refinement,
where the model updates its parameters in each iteration. The process ensures that the model learns
not only from correct code solutions but also from common mistakes, thereby improving its ability to
generate high-quality code across diverse tasks. This iterative refinement, guided by self-generated
comparative data, distinguishes CODELUTRA from traditional supervised fine-tuning approaches.

---

**Algorithm 1:** CODELUTRA

---

**Input** : Training set $\mathcal{D} = \{(x_i, y_i)\}_{i=1}^n$; Initial base model $\pi_0$; Number of code responses per
input $M$; Number of preference pairs per input $K$; Number of iterations $T$;
Hyperparameter $\lambda$.

**for** $t = 0$ **to** $T - 1$ **do**
    Initialize preference dataset $\mathcal{D}_t = \emptyset$;
    ForEach $x_i \in \mathcal{D}$ Initialize chosen code set $Y_i^{(c)} = \emptyset$;
    Initialize rejected code set $Y_i^{(r)} = \emptyset$;
    **for** $k = 1$ **to** $M$ **do**
        Generate response $\hat{y}_i^k \sim \pi_t(x_i)$;
        **if** *Execution result of $\hat{y}_i^k$ matches ground truth $y_i$* **then**
            Add $\hat{y}_i^k$ to $Y_i^{(c)}$;
        **else**
            Add $\hat{y}_i^k$ to $Y_i^{(r)}$;

    **for** $k = 1$ **to** $K$ **do**
        Randomly sample $\hat{y}_i^{c_k}$ from $Y_i^{(c)}$ (with replacement if $|Y_i^{(c)}| < K$);
        Randomly sample $\hat{y}_i^{r_k}$ from $Y_i^{(r)}$ (with replacement if $|Y_i^{(r)}| < K$);
        Add $(x_i, \hat{y}_i^{c_k}, \hat{y}_i^{r_k})$ to $\mathcal{D}_t$;
    Update model $\pi_{t+1}$ by minimizing the combined loss:

$$\pi_{t+1} = \operatorname{argmin}_{\pi_\theta} \left[ \underbrace{-\mathbb{E}_{(x_i, y_i^c, y_i^r) \sim \mathcal{D}_t} \left[ \log \sigma \left( \beta \left( \log \frac{\pi_\theta(y_i^c|x_i)}{\pi_t(y_i^c|x_i)} - \log \frac{\pi_\theta(y_i^r|x_i)}{\pi_t(y_i^r|x_i)} \right) \right) \right]}_{\text{compare correct and incorrect solutions}} \right.$$

$$\left. - \underbrace{\lambda \mathbb{E}_{(x_i, y_i^c) \sim \mathcal{D}_t} \left( \log \pi_\theta(y_i^c|x_i) \right)}_{\text{maximize likelihood for correct codes}} \right], (7)$$

---

### A.2   EXAMPLES FOR DIFFERENT DATASETS.

We provide examples that highlight the tasks used to evaluate our framework. The first example
illustrates the Data Query task (see Figure 6), where models generate SQL queries from natural
language descriptions based on a given database schema. The second example showcases the Data
science task (see Figure 7), in which models write Python code to solve typical data manipulation
problems, such as processing a data frame. These examples reflect common real-world applications
of language models in both querying databases and performing data science operations.

---

**Different answers from different models for the data query:**

**Database schema:**

```
CREATE TABLE customers ( CustomerID INTEGER  UNIQUE not null

primary key, Segment TEXT null, Currency TEXT null );

CREATE TABLE gasstations ( GasStationID INTEGER UNIQUE not null primary key,

ChainID INTEGER null, Country TEXT null, Segment TEXT null );

(Omit other database information...)
```

**Ground truth another:**

```
SELECT T2.Consumption  FROM transactions_1k AS T1
INNER JOIN yearmonth AS T2 ON T1.CustomerID = T2.CustomerID
WHERE T1.Price / T1.Amount > 29.00
AND T1.ProductID = 5   AND T2.Date = '201208';
```

Figure 6: An example from the data query dataset from the BIRD (Wang et al., 2023a).

---

**An example of data science from the DS1000 (Lai et al., 2023):**

**Problem:**

```
I have a simple dataframe which I would like to bin for every 4 rows.

It looks like this:

col1\n0    1\n1    1\n2    4\n3    5\n4    1\n5    4\n

and I would like to turn it into this:

col1\n0    11\n1    5\n

I have already posted a similar question here

but I have no idea how to port the solution to my current use case.

Can you help me out?
```

**Solution:**

```
def g(df):
    return df.groupby(df.index // 4).sum()

result = g(df.copy())
```

Figure 7: An example of data science from the DS1000 (Lai et al., 2023).

## A.3 MORE RESULTS

Models trained with the DPO loss are capable of assessing the quality of code answers. To prevent data leakage, we utilized the robust open-source model Codestral to generate multiple samples on Bird's test set, constructing positive and negative sample pairs based on execution accuracy. We evaluated the fine-tuned LLM's ability to accurately assess code quality by measuring the classification accuracy on this dataset. Under the standard supervised fine-tuning (SFT) setting, the model achieved a classification accuracy of 56%, which is close to random guessing and indicates that SFT alone lacks this capability. In contrast, our CODELUTRA attain a classification accuracy of 79%, demonstrating that

Table 4: Code quality assessment accuracy.

| Methods | Accuracy (%) |
|---|---|
| Supervised fine-tuning | 56.3 |
| Preference learning | 79.6 |

our approach enables the model to better understand code characteristics and select correct answers. This substantial improvement highlights the significant potential of CODELUTRA.

## A.4 EXPERIMENTAL SETUP

Table 5 summarizes the training hyperparameters used for data query and data science tasks across each iteration. It includes key training parameters such as learning rate, batch size, LoRA rank, etc.

Table 5: Summary of training hyperparameters for data query and data science for each iteration.

|  | Parameters | Value |
|---|---|---|
| Data query | Number of epochs | 1 |
|  | Learning rate | $5 \times 10^{-5}$ |
|  | $\beta$ | 0.1 |
|  | Batch size | 16 |
|  | Gradient accumulation steps | 1 |
|  | Maximum sequence length | 2048 |
|  | DeepSpeed Zero stage | 2 |
|  | Weight decay | 0.0001 |
|  | LoRA rank | 8 |
|  | $\lambda$ | 1.0 |
| Data science | Number of epochs | 1 |
|  | Learning rate | $5 \times 10^{-5}$ |
|  | $\beta$ | 0.5 |
|  | Batch size | 16 |
|  | Gradient accumulation steps | 1 |
|  | Maximum sequence length | 512 |
|  | DeepSpeed Zero stage | 2 |
|  | Weight decay | 0.0001 |
|  | LoRA rank | 8 |
|  | $\lambda$ | 0.5 |

We set $K$=10 for each iteration, generating 10 positive and negative sample pairs per question. To maintain quality when selecting incorrect samples, we filter out answers that contain repeated strings.

## B LIMITATION AND FUTURE WORK

While CODELUTRA significantly enhances code generation performance by leveraging self-generated comparative data, it exhibits several limitations that warrant consideration. The current framework primarily focuses on the correctness of the generated code, overlooking other vital aspects such as efficiency, readability, and adherence to specific formal specifications, which are essential for practical applications. Additionally, CODELUTRA treats all failed code attempts uniformly, without distinguishing between different types or severities of errors, potentially limiting the model's ability to learn from more informative mistakes.

To address the aforementioned limitations, future research related to CODELUTRA should explore several key directions. Expanding the preference-guided refinement mechanism to incorporate additional criteria such as code efficiency, readability, and compliance with formal specifications would enhance the overall quality and utility of the generated code. Developing a more nuanced approach to categorizing and prioritizing failed code attempts based on the type and severity of errors could enable more targeted and effective learning, thereby improving the model's ability to avoid similar mistakes in future generations. Exploring alternative evaluation methods, such as static code analysis or formal verification tools, could reduce the framework's reliance on execution results and broaden its applicability to a wider range of tasks.

