# OpenReview forum: "CodeLutra: Boosting LLM Code Generation via Preference-Guided Refinement"
_ICLR.cc/2025/Conference — ICLR 2025 Conference Withdrawn Submission_

### Official Review · Reviewer_Ur5D · 2024-10-27

**Soundness:** 3
**Presentation:** 3
**Contribution:** 3
**Rating:** 6
**Confidence:** 3

**Summary:**

The authors proposed a new training framework called CODELUTRA which aims to fine-tune a small CodeLM to match or surpass the closed-source LLMs like GPT-4. CODELUTRA adopts an iterative method to learn by comparing the correct generation and the failed generation. At each iteration, CODELUTRA constructs the preference dataset by classifying the generation codes of the model from the last iteration and employs a dual-loss function that combines DPO with SFT for training. The authors show that their method can achieve a performance comparable to GPT-4 in the data query and data science tasks.

**Strengths:**

1. The paper is well-organized and easy to follow
2. The proposed method can lead to a fine-tuned LLAMA3-8B model which has comparable performance to GPT-4.
3. The authors conduct comprehensive ablation studies that the effect of every component involved in their method is clearly demonstrated.
4. The method can still have good performance with limited annotations or training samples.

**Weaknesses:**

1. Line 230 states that "The refinement process continues until the improvement between consecutive iteration becomes marginal". However, in the experiments, the authors seem to fix the iteration number to 4. In practice, how do you decide if the improvement between consecutive iterations is marginal?

2. The baseline setup is not clear enough and may not be comprehensive.
a) For closed-source LLMs, it is unknown what prompting method is used. It is also not clearly stated what fine-tuning method is used. From the Appendix, I infer that the LoRA is used in CODELUTRA but is it also used in the fine-tuning baseline?
b) In Table 1, since LLAMA-3 is used as the base model for CODELUTRA, the authors should apply more previous fine-tuning methods in the same setting and compare with them instead of comparing with different open-source CodeLLMs. For example, the related work section mentions other fine-tuning methods (Line 520), e.g. Self-debug and Codefort. The authors should apply them to fine-tune LLAMA3 and compare the results.

3. Paper presentations can be further improved. Specifically, a) Line 142 $f$ is not defined. b) The notations in the legend of Figure 2 are not defined.

**Questions:**

1. See weakness 1, 2
2. I am curious that if this method can lead to a model that is generalizable. For example, the authors split DS-1000 for training and evaluation. I wonder how the resulting model would perform on other similar datasets, e.g. MBPP?

---

> ### Author Response · Authors · 2024-11-22
> **Response to Reviewer Ur5D**
>
> > **W1: how to decide if the improvement between consecutive iterations is marginal.**
>
> Thank you for pointing out this discrepancy. In practice, we set a threshold parameter, $\mu$, to determine when the improvement between consecutive iterations becomes marginal. Specifically, if the accuracy improvement between two iterations is less than $\mu$ (we set $\mu = 0.5$ in our experiments), the refinement process terminates. For simplicity and consistency in our current experiments, we fixed the number of iterations to 4, as this was sufficient to achieve convergence on the benchmarks we evaluated. We will update the corresponding algorithm box and revise the description to clearly reflect this threshold-based stopping criterion for greater accuracy.
>
> > **W2: The baseline setup is not clear enough and may not be comprehensive.**
>
>
> Thank you for raising these points. We appreciate the opportunity to clarify and enhance the description of our baseline setup while incorporating additional experimental results for better comparison:
>
> - **(a) Fine-tuning and prompting Methodology of reported results**: We confirm that all SFT experiments, including those in CodeLutra, are conducted using LoRA as the fine-tuning method. Regarding prompting, as detailed in Lines 255–266, we adopt a standard prompt format that includes only background information and a natural language problem description. No additional information, such as execution results or reference answers, is provided during training or evaluation. This consistent prompt format ensures a fair and unbiased comparison across all methods, and all reported results are based on this standardized setup.
> - **(b) Comparison with previous fine-tuning methods**: We acknowledge the reviewer’s suggestion to include more fine-tuning methods applied to LLAMA-3 for a direct comparison. To address this, we have conducted additional experiments, incorporating Self-debug and Codefort under the same settings. Below, we present the updated results for the Spider benchmark using LLAMA-3-8B as the base model:
>
> | **Methods**          | **Execution Accuracy** |
> |-----------------------|-------------------------|
> | SFT                  | 67.9%                  |
> | SFT + Self-debug     | 69.2%                  |
> | CodeLutra (Ours)     | 76.3%                  |
>
> These results highlight that while Self-debug offers a marginal improvement over standard SFT, CodeLutra achieves significantly higher execution accuracy, demonstrating its effectiveness in leveraging both successful and failed generations iteratively to refine model performance.
>
> We will update the manuscript to incorporate these clarifications and results. These additions not only provide a more comprehensive comparison but also reinforce the advantages of CodeLutra in the context of fine-tuning for code generation tasks. Thank you again for your valuable feedback.
>
> > **W3:The missing defination of notation.**
>
> Thank you for highlighting these issues. We agree that clearer definitions and notations would improve the paper's presentation.
>
> > **Q2: Evaluation on more tasks.**
>
> **A**: Thank you for your question regarding the generalizability of our method. To evaluate this, we tested CodeLutra on the MBPP dataset from EvalPlus [1] using Llama-3-8B as the base model. The model was trained with 3 iterations, and the results are as follows:
>
> | **Methods**          | **Accuracy** |
> |-----------------------|-------------------------|
> | Base                  | 52.3%                  |
> | SFT                   | 53.1%                  |
> | CodeLutra             | 55.4%                  |
>
> The results demonstrate that CodeLutra generalizes well to MBPP, achieving notable improvements over both the base model and the SFT baseline. This suggests that our approach is effective not only on DS-1000 but also on other datasets, highlighting its generalizability and applicability to diverse code generation tasks.
>
> [1] Liu, Jiawei, et al. "Is your code generated by chatgpt really correct? Rigorous evaluation of large language models for code generation." Advances in Neural Information Processing Systems 36 (2024).

---

> > ### Comment · Reviewer_Ur5D · 2024-11-25
> >
> > Thanks for your response!
> > Two follow up questions:
> > 1. Regarding the finetuning setup, why not use & compare with the full-parameter finetuning?
> > 2. Since your method requires finetuning data, I think close-source LLM should be compared with under the in-context learning setting, e.g. providing demonstration examples.

---

### Official Review · Reviewer_yWax · 2024-10-28

**Soundness:** 2
**Presentation:** 3
**Contribution:** 2
**Rating:** 3
**Confidence:** 4

**Summary:**

The paper proposes CODELUTRA, a preference-guided training framework to let code LLMs iteratively refine itself based on execution signals from its own generations. Specifically, given a task-specific training set, at each iteration, the model generates answers which are then evaluated by unit tests. Each correct answer is paired with an incorrect answer to form a preference pair. The preference dataset is then used for DPO training. To address the issue that DPO may reduce the generation probability of both correct and incorrect answers, supervised finetuning loss is added to DPO loss for joint training. Experiments show that CODELUTRA significantly improves performance on SQL and data science tasks, and is much more effective than DPO alone.

**Strengths:**

* Comprehensive evaluation, ablation, and analysis support the effectiveness of the proposed method. In particular, the necessity of negative training samples and of SFT loss are both well studied.
* The paper is well written and easy to follow.

**Weaknesses:**

* The technical novelty of this paper is somewhat limited. L233-246 claimed two major points of novelty: refinement from execution feedback and dual loss mechanism. First, using feedbacks from program execution to iteratively refine code LLMs is a direction that has been extensively studied (e.g., CodeRL [1], and NExT [2]). However, these works are not discussed in the related work section. Second, the dual loss objective (i.e. adding SFT loss in DPO training) was proposed in [3], known as RPO, which is not cited.
* I find DS-1000 Pass@1 results in Table 1 are inconsistent with the public leaderboard (https://ds1000-code-gen.github.io/model_DS1000.html). In particular, pass@1 of Codestral-22B and Llama-3-70B-Chat is 51.2 and 48.6 respectively in the leaderboard, but 35.8 and 36.4 respectively as reported in the paper.


Reference:

[1] Le, Hung, et al. "Coderl: Mastering code generation through pretrained models and deep reinforcement learning." Advances in Neural Information Processing Systems 35 (2022): 21314-21328.

[2] Ni, Ansong, et al. "NExT: Teaching Large Language Models to Reason about Code Execution." Forty-first International Conference on Machine Learning.

[3] Liu, Zhihan, et al. "Provably mitigating overoptimization in rlhf: Your sft loss is implicitly an adversarial regularizer." arXiv preprint arXiv:2405.16436 (2024).

**Questions:**

* Are the two answers in Figure 5 flipped? Given Currency is in table customers, I feel the first answer is correct, and the second is wrong.

---

> ### Author Response · Authors · 2024-11-22
> **Response to Reviewer yWax**
>
> > W1: The novelty of this paper
>
> Although both our method and previous work [1,2] utilize execution results, our high-level goals are fundamentally distinct. Previous approaches used execution results primarily to **refine incorrect code until arriving at the final correct solution**. In contrast, we leverage execution results to help the model **distinguish between correct and incorrect outputs via the Bradley–Terry model, maximizing the likelihood difference between them**. Our iterative process enables the model to gradually improve its performance. Our framework is simpler and more straightforward compared to prior methods.
>
> For instance, **CodeRL** [1] requires a complex reward mechanism and the training of a critic model to continuously optimize the actor model. **NExT** [2] depends on prompt engineering to generate chain-of-thought rationales for solving coding tasks, which is feasible only for advanced models (e.g., GPT-4). In contrast, our goal is to train open-source LLMs to match or even outperform closed-source models using a simple  framework. Below, we outline the key distinctions between our method and previous code generation approaches that utilize execution results:
>
> 1. **Simplicity in using execution results**:
>    Unlike prior methods, we do not differentiate between specific failure reasons. Instead, we pair successful attempts with failed ones without introducing any prior information, allowing the model to discern differences through the Bradley–Terry model. This enables the model to iteratively improve failed attempts during training (see Figure 4(b)). By doing so, we avoid the complexities of reward design and the challenges posed by distribution shifts that may hinder generalization.
>
> 2. **Independence from complex prompts**:
>    While complex prompts can enhance model performance, they come with significant drawbacks:
>    - **Applicability to small models**: Smaller models struggle to comprehend advanced instructions, making prompt-based methods unsuitable for them.
>    - **Computational overhead**: Prompt-based methods often require additional guidance or multi-turn interactions, which increase computational costs.
>    In contrast, our approach relies only on basic, minimal prompts. Through efficient fine-tuning, we enable smaller models to achieve—or even surpass—the performance of larger models, eliminating the reliance on complex prompting strategies.
>
> 3. **Effectiveness without Unit Tests or ground truth**:
>    Previous methods, such as CodeRL and NExT, rely heavily on Unit Tests or ground truth to validate code correctness. Our method, however, can deliver strong performance by simply assessing whether the code is executable (see Figure 3c). This demonstrates the robustness and generalization capability of our approach, even in scenarios where ground truth is unavailable.
>
> Thank you for pointing this out. We were unaware of the existence of [3] at the time of submission. While [3] focuses on the theoretical explanation of the dual loss objective, our work primarily emphasizes how this approach is adapted, refined, and applied to a fundamentally different domain—code generation—which presents unique challenges such as strict syntactic and semantic correctness, leveraging execution feedback, and iterative refinement. We believe this distinction highlights the novelty and contribution of our work. In the next revision, we will add a proper citation to [3] and revise the relevant descriptions to acknowledge its contributions accurately. Thank you for bringing this to our attention.
>
>
> > **W2: Inconsistent with the public leaderboard**
>
> Thank you for pointing this out. We had also noticed the discrepancy between our reported results and the public leaderboard. **To ensure fairness in comparison, we used identical sampling parameters and prompts across all models without any specific tuning for these two models**. However, we will re-evaluate the models using the exact settings reported on the public leaderboard to ensure the reliability of the reported results. We appreciate your feedback and will update the manuscript accordingly to reflect the findings.
>
>
> [1] Le, Hung, et al. "Coderl: Mastering code generation through pretrained models and deep reinforcement learning." Advances in Neural Information Processing Systems
> [2] Ni, Ansong, et al. "NExT: Teaching Large Language Models to Reason about Code Execution." Forty-first International Conference on Machine Learning.
> [3] Liu, Zhihan, et al. "Provably mitigating overoptimization in rlhf: Your sft loss is implicitly an adversarial regularizer." arXiv preprint arXiv:2405.16436 (2024).
>
> > **Q1: Are the two answers in Figure 5 flipped?**
>
> Thank you for catching this mistake. You are correct that the two answers in Figure 5 are flipped. Given that Currency is in the table customers, the first answer is correct, and the second is incorrect. We will revise Figure 5 accordingly. Thank you for bringing this to our attention.

---

> > ### Comment · Reviewer_yWax · 2024-11-25
> >
> > Thank you for the response. I'm inclined towards keeping the initial score, with reasons detailed below.
> > > Previous approaches used execution results primarily to refine incorrect code until arriving at the final correct solution. In contrast, we leverage execution results to help the model distinguish between correct and incorrect outputs via the Bradley–Terry model, maximizing the likelihood difference between them.
> >
> > I would regard this as primarily the difference between RLHF (or RLEF where E=Execution) and DPO, which has been studied albeit in text domain. The paper does not present a comparative study of these two approaches for code.
> >
> > > Simplicity in using execution results & Independence from complex prompts
> >
> > I believe simplicity is a strength only if one can show the simple method is more or equally effective than other more complicated baselines. The paper lacks such comparative results.
> >
> > > Effectiveness without Unit Tests or ground truth
> >
> > PPOCoder [4] also used executability as reward signal (see their 3.2.1). The general idea of CodeLutra is also somewhat similar to CodePPO except for using DPO rather than PPO.
> >
> > > While [3] focuses on the theoretical explanation of the dual loss objective, our work primarily emphasizes how this approach is adapted, refined, and applied to a fundamentally different domain—code generation—which presents unique challenges such as strict syntactic and semantic correctness, leveraging execution feedback, and iterative refinement.
> >
> > The adaptation seems trivial as it is simply to add an SFT loss to the DPO objective. I don't see how the unique challenges mentioned here are related to the adaptation.
> >
> > In general, the absence of a comprehensive literature review is a major weakness of this paper. The current draft is presented as if CodeLutra were the first to leverage execution feedback for aligning CodeLLMs, which is not the case. I encourage the authors to revise the paper by including relevant works and using them as baselines in experiments wherever necessary.
> >
> > **Reference**
> >
> > [4] Shojaee, Parshin, et al. "Execution-based Code Generation using Deep Reinforcement Learning." Transactions on Machine Learning Research.

---

### Official Review · Reviewer_6tvY · 2024-11-03

**Soundness:** 2
**Presentation:** 2
**Contribution:** 1
**Rating:** 3
**Confidence:** 4

**Summary:**

The paper introduces CODELUTRA, a framework designed to enhance the performance of LLMs in code generation tasks. However, the method is almost the same as an existing method.

**Strengths:**

NA

**Weaknesses:**

1. The proposed method closely resembles that presented in [1]. Applying the same approach to a different scenario does not warrant publication, especially since this new scenario is simpler and benefits from execution feedback.

[1] Iterative Reasoning Preference Optimization. https://arxiv.org/abs/2404.19733

**Questions:**

NA

---

> ### Author Response · Authors · 2024-11-22
> **Response to Reviewer 6tvY**
>
> > W1: The novelty of this paper
>
>
> Thank you for your feedback on our submission. We belive **novelty lies in how the approach is adapted, refined, and applied to a fundamentally different domain with unique challenges**. In a simple analogy, we should not deny the novelty of LLM papers simply because they all rely on transformers as their foundational technique. Below we highlight our distinct contributions:
>
>
> 1. As stated in Lines 145–156, **code generation is not merely a "different scenario" but a domain with distinct requirements and challenges**, such as execution correctness, syntax, and logical consistency. Prior work on mathematical reasoning has not tackled these unique aspects.  For code generation, even minor errors, such as missing tokens or incorrect syntax, can render the entire code non-functional, making the task significantly more sensitive and complex than reasoning tasks. **It is important to note that method that works for mathematical reasoning does not necessarily transfer to code generation**. For example, simply using DPO shows poor results (see Table 2 in our work), while using DPO show satisfactory performance for reasoning task. Our paper demonstrates phenomena that are completely different in code generation compared to other tasks, which will have strong guiding and reference significance for the subsequent research of the community.
>
> 2. We would like to clarify that **CodeLutra focuses on leveraging the correctness of generated answers rather than the feedback information itself**. This distinguishes our work from previous methods that often rely on directly incorporating feedback information (e.g., error messages or debugging hints) to guide model improvements. Our approach avoids overfitting to specific error patterns and ensures broader generalizability by iteratively refining outputs using both successful and failed generations. We believe this is novel.
>
> 2. **In-depth understanding**: It is also important to understand a method's performance, generalizability, and limitations under different scenarios. Our method goes beyond simply applying [1]’s framework to a new scenario. Section 4 and further analyses highlight CodeLutra’s insights unique to code generation that are not presented in [1]:
>    - **Failed Attempts Matter**: Incorporating failed attempts in training leads to significant performance improvements in code generation, achieving results comparable to or exceeding leading models.
>    - **Applicability without ground truth**: Even without high-quality annotations, using executability as a metric enables effective refinement, reducing common errors like syntax issues, which remarkably improve the preformance of code generation. **This is not explored in reasoning task**.
>     - **Improve the incorrect answers**: Further analysis shows that CodeLutra improves the quality of incorrect answers across iterations. Specifically, the similarity between the model’s outputs on the error set and the ground truth increases significantly across iterations, as illustrated in Figure 3(b). At the same time, the similarity for correct answers remains stable, ensuring that improvements focus on error cases without degrading already accurate outputs.
>
> 3. **Results demonstrating real-world impact**. Our framework achieves state-of-the-art results, matching or surpassing GPT-4 on specific metrics, using only limited training data. This demonstrates the practical and scientific significance of CODELUTRA in democratizing access to high-performing LLMs for code generation tasks.
>
>
> These distinctions emphasize that CodeLutra addresses the specific challenges of code generation with a principled and scalable approach, making it fundamentally different and more specialized than previous work.
>
> [1] Pang, Richard Yuanzhe, et al. "Iterative reasoning preference optimization." arXiv preprint arXiv:2404.19733 (2024).

---

> ### Comment · Reviewer_6tvY · 2024-11-25
>
> Thanks for the authors' response.
>
> First, the authors acknowledge that their method is fundamentally the same as the approach described in [1]. For instance, the iterative DPO strategy and the second loss term in Equation 6 were already proposed in [1]. However, the paper does not appropriately credit [1] for these contributions. Instead, it appears that this connection is deliberately obscured, which is unacceptable.
>
> Second, the use of execution feedback is not a novel contribution, as this is a well-established technique in the domain of code generation.
>
> Finally, while the code domain differs from mathematics, I do not perceive a fundamental distinction between the two, as many techniques can be applied to both domains. In fact, this work itself demonstrates that overlap. However, I do not find valuable insights contributed by this work.
>
> [1] Iterative Reasoning Preference Optimization. https://arxiv.org/abs/2404.19733

---

### Official Review · Reviewer_RkM7 · 2024-11-03

**Soundness:** 4
**Presentation:** 4
**Contribution:** 3
**Rating:** 5
**Confidence:** 4

**Summary:**

This paper proposes a method by iteratively generating successful and failed code and training with preference optimization.

**Strengths:**

The paper is well-written. The proposed method with training with correct and failed generations iteratively makes sense. Experiments show good improvement on benchmarks.

**Weaknesses:**

Some experimental setup is not clear enough, such as training data, SFT setting, and details of synthetically generated dataset. One of the contribution DPO and SFT loss is studied in previous literature. More experiments might be needed for comparing SFT then DPO with DPO+SFT loss.

**Questions:**

- In line 150, "if the model only predicts wrongly in the final token in a code snippet, the overall probability P (y|x) in the Equation 1 might still remain high as the preceding tokens are correct". While the hypothesis makes sense, do you really observe this situation in real LLM and dataset? I doubt it.
- Equation 6 is studied in a previous literature "Provably Mitigating Overoptimization in RLHF: Your SFT Loss is Implicitly an Adversarial Regularizer" with theoretical support, but not cited. This also limits the novelty contribution (at least on this part).
- I'm confused with the experimental setup. What is the training dataset? It seems the experiment is using the test dataset to train. Could you clarify?
- Could you explain the setting for SFT in Table 1? One baseline is the SFT model that only uses the groundtruth training solutions, or use the synthetically generated correct solutions. Which one are you using?
- I don't think Table 2 is a right setting, where 17.2 and 12.4 is extremely low for DPO-only method. Normally we do DPO training on top of SFT model. The right setup should be training on top of SFT model. What is the gap between, SFT then DPO training and the SFT regularized preference training?
- 500 samples might mean that 500 prompts or problems. What is the size of generated samples overall?

---

> ### Author Response · Authors · 2024-11-22
> **Response to Reviewer RkM7**
>
> >  W1: where a model predicts the final token incorrectly but maintains a high overall probability due to preceding correct tokens—actually occur in real LLMs and datasets?
>
> Thank you for your question! Your skepticism is completely reasonable as such a tricky case could not always happen. The statement in our paper is not based on statistical validation of specific data distributions but rather serves as a subjective explanation intended to help readers better understand how errors in code generation might be perceived.
>
> Specifically, we use this example to illustrate a key characteristic of code generation tasks: even if the final generated code does not meet expectations (e.g., an error in the last token leads to syntax or functionality issues), the probability $P(y|x)$ assigned by the model might still be high because preceding tokens are correct. This phenomenon highlights a potential blind spot in traditional likelihood-based methods when assessing code generation quality. While we did not directly analyze the frequency of such cases in real datasets or models, this "locally correct but globally incorrect" issue is a common challenge in code generation. Our explanation aims to conceptually clarify why more sophisticated training and evaluation methods, such as our CodeLutra framework, are necessary. CodeLutra addresses this by leveraging both negative samples and failed generations to better model these error patterns. In future work, we can further analyze the distribution of different error types and their relationship with model probabilities to provide more empirical support for this hypothesis. Thank you for your valuable suggestion!
>
> > W2: missing citation of the paper and novelty.
>
> Thank you for pointing out the connection between Equation 6 and the prior work "Provably Mitigating Overoptimization in RLHF: Your SFT Loss is Implicitly an Adversarial Regularizer." We were unaware of this paper during submission and will make sure to include it in the revised manuscript.
>
> While the prior work provides theoretical insights, our contribution focuses on a practical and scalable framework for iterative refinement in code generation, which has not been explored in prior literature. Specifically, we empirically demonstrate the effectiveness across multiple benchmarks and model families, integrate it with dual loss and negative samples, and showcase its role in stabilizing performance gains across iterations. This strengthens the theoretical foundation of our work while highlighting its practical novelty.
>
> > **W3: What is the training dataset?**
>
> We describe our training dataset construction in **Lines 185-200**. The training corpus is constructed purely based on the training split of each benchmark. **We do not use test set for training**. Take DS-1000 as an example, we split it into 500 samples for training and 500 for evaluation. For each of those 500 training examples, we construct 10 preference pairs by sampling multiple solutions from current policy. The pairs are then formed by selecting one correct solution and one incorrect solution.
>
> > **W4: The setup of the SFT in Table 1.**
>
> All the SFT results we report are obtained by fine-tuning one epoch on the ground truth training solutions. We will clarify this further in the paper.
>
> > **W5: DPO experiment**
>
> That’s a great question! We do use the standard DPO training practice you suggested----DPO training is indeed performed on top of SFT model.
>
> In our paper, we provided further insight and explanation on the suboptimality of DPO in **L201-L213** and **Figure 2**. DPO presents a notable limitation due to its tendency to decrease the likelihood of correct code during training. This is evidenced by the dashed lines in Figure 2, which is also observed in prior works. This diminishing likelihood can negatively affect code generation, especially since  correct code are critical for successfully solving the assigned tasks. It is crucial, therefore, to prioritize the likelihood of correct solutions to the task at hand.
>
> In practice, we observe that when the DPO model converges, the model's answers often output some repeated words, resulting in completely unexecutable code. We provide an example below on Spider, trained with DPO (on top of SFT). The output of the model is:
>
> > Citizenship AS \"Citizenship\" FROM (SELECT (SELECT Citizenship AS Citizenship(SELECT(SELECT(SELECT(SELECT(SELECT(SELECT(SELECT(SELECT(SELECT...... （many repeated '(SELECT' ）
>
> > **W6: Generated samples number for training**
>
> As shown in **L950**, in each iteration, we set the number of positive and negative sample pairs corresponding to each question to 10. Therefore, 500 questions in each iteration correspond to 5000 training samples.

---

### Official Review · Reviewer_kALm · 2024-11-04

**Soundness:** 3
**Presentation:** 3
**Contribution:** 3
**Rating:** 8
**Confidence:** 4

**Summary:**

This paper presents CodeLutra, a supervised fine-tuning (SFT) approach that demonstrates significant improvements on coding tasks. Specifically, CodeLutra achieves GPT-4-level performance fine-tuning an open-source 8-billion-parameter model using as few as 500 samples (with groundtruth solutions). The key idea behind CodeLutra is to use both positive and negative examples to fine-tune the model, creating a hybrid of SFT and DPO techniques. CodeLutra assumes a ground truth to generate these examples: if a sample code produces the same inputs/outputs as the ground truth, it is a positive example; otherwise, it is a negative example.

**Strengths:**

- CodeLutra simple yet effective method, with clear articulation of how it differs from related work.

- Impressive results: with only 500 samples, CodeLutra achieves GPT-4-level performance on a base model with just 8 billion parameters. For the Spider benchmark, it improves base model performance from 59.3 to 74.4 in just four iterations, surpassing GPT-4’s 74.4. On BIRD, it increases performance from 22.3 to 42.6 in four iterations, approaching GPT-4’s 46.3.

- Comprehensive evaluation, covering three coding benchmarks (Spider, BIRD, and DS-1000) and three models (Llama-3-8B, Gemma-7B, and StarCode-7B), demonstrating the approach's generalizability.

- Strong ablations that address key questions: (i) dual loss significantly boosts performance, raising it from 17.2 (DPO) to 76.6 on Spider; (ii) negative samples are crucial, as performance increases from 20 to over 40 with their inclusion, while positive samples alone yield minimal improvement.

**Weaknesses:**

- Current evaluation focuses on SQL queries and data science problems, which are relatively short (from a few lines of code to several 10s of lines of code). It would be interesting to see how this approach generalizes to longer programs.
- Limited exploration of scenarios without ground truth. In such cases, CodeLutra relies on syntactic error detection, but the results are, as expected, less impressive.

**Questions:**

- How does CodeLutra perform on longer programs, e.g,, competitive coding? (This is nice to have; the material in the paper is enough for publication.)
- What is the CodeLutra's performance vs program length on the current data sets?

---

> ### Author Response · Authors · 2024-11-22
> **Response to Reviewer kALm**
>
> > **W1/Q1: how this approach generalizes to longer programs.**
>
> We acknowledge that the current work does not evaluate CodeLutra on longer programs, such as those in competitive coding scenarios. This limitation arises from the lack of publicly available, well-benchmarked datasets for long and complex programs. Developing or identifying such datasets is an important direction for future research. That said, we believe the iterative refinement approach of CodeLutra, combined with execution feedback, has the potential to generalize to longer programs, though challenges such as error propagation and increased complexity would need to be carefully addressed.
> > **W2: Exploration of scenarios without ground truth.**
>
> Thank you for raising this point. While CodeLutra relies on ground truth for preference dataset collection, we tested its performance using executability as the metric instead. On the BIRD dataset, execution accuracy improved from 22.3% to 30.9%, and the proportion of executable code increased from 59.8% to 89.7%. This shows that even without ground truth, the model effectively avoids common errors and achieves substantial improvements. These results highlight CodeLutra's robustness and applicability in scenarios without high-quality annotations.
>
> > **Q2: CodeLutra's performance vs program length on the current data sets.**
>
> We evaluate CodeLutra's performance across different program lengths by categorizing the generated code based on the number of lines and measuring the execution accuracy within each category. The results are as follows:
>
> | **Length (lines)** | **Execution Accuracy** |
> |---------------------|-------------------------|
> | 0–5                | 48.7%                  |
> | 5–10               | 34.8%                  |
> | 10+                | 1.7%                   |
>
> From these results, we observe that CodeLutra performs best on shorter programs (0–5 lines), where the simplicity of the code enables the iterative refinement process to achieve high accuracy. For medium-length programs (5–10 lines), accuracy decreases as the complexity increases, indicating that longer dependencies and errors become more challenging to address. This analysis highlights the current limitations of CodeLutra for handling longer programs, suggesting the need for additional refinement techniques and better training data to address these challenges. We will include this detailed analysis in the revised manuscript to provide further insights into CodeLutra's performance across different code lengths.

---

### Author Response · Authors · 2024-11-23
**General response**

We sincerely thank all the reviewers for their time and insightful feedback. We are encouraged that reviewers recognize it as _simple yet effective_, with a _clear articulation of its novelty_ (kALm, RkM7, yWax). Reviewers appreciated the _impressive results achieved by CodeLutra_, as well as the _comprehensive evaluation, ablations, and analysis_ demonstrating its _generalizability and effectiveness_ (kALm, yWax, RkM7). Furthermore, reviewers commended the _clarity and organization of our paper_ (RkM7, yWax).

The importance of our work is underscored by its impressive effectiveness in code generation. Below, we summarize the key contributions of our work:

1. **A simple yet effective approach**: CodeLutra introduces an iterative training framework that leverages both correct and failed generations to refine LLM performance. This approach significantly differentiates itself from existing methods and demonstrates **state-of-the-art results** on various benchmarks.
2. **CodeLutra achieves GPT-4-level performance** on a base model with 8B parameters, with only 500 samples. On the Spider benchmark, CodeLutra improves performance from 59.3 to 76.6, surpassing GPT-4’s 74.4.
3. **Comprehensive evaluation and generalizability**: Our study evaluates CodeLutra across three challenging coding benchmarks (Spider, BIRD, and DS-1000) and diverse model families (Llama-3-8B, Gemma-7B, and StarCode-7B), showcasing the robustness and generalizability of the proposed method.
4. **Strong ablation studies to explain why the method works**: We present detailed analyses answering key questions, such as the necessity of dual loss (raising performance from 17.2 to 76.6 on Spider) and the importance of negative samples.

We address each reviewer's comments in detail below, incorporating additional experiments and analyses as suggested.

---

### Note · Authors · 2024-11-27

I have read and agree with the venue's withdrawal policy on behalf of myself and my co-authors.